# Extent of Cardiac Damage and Mortality in Patients Undergoing Transcatheter Aortic Valve Implantation

**DOI:** 10.3390/jcm10194563

**Published:** 2021-09-30

**Authors:** Marisa Avvedimento, Anna Franzone, Attilio Leone, Raffaele Piccolo, Domenico Simone Castiello, Federica Ilardi, Andrea Mariani, Roberta Esposito, Cristina Iapicca, Domenico Angellotti, Maria Scalamogna, Ciro Santoro, Luigi Di Serafino, Plinio Cirillo, Giovanni Esposito

**Affiliations:** Department of Advanced Biomedical Sciences, University of Naples Federico II, Via Sergio Pansini 5, 80131 Naples, Italy; m.avvedimento@gmail.com (M.A.); anna.franzone@unina.it (A.F.); attilio.leone8@gmail.com (A.L.); raffaele.piccolo@unina.it (R.P.); ds.castiello@gmail.com (D.S.C.); fedeilardi@gmail.com (F.I.); dr.marianiandrea@gmail.com (A.M.); roberta.esposito1@unina.it (R.E.); cristinaiapicca@gmail.com (C.I.); dom.angellotti@gmail.com (D.A.); maria.scalamogna@libero.it (M.S.); ciro.santoro@unina.it (C.S.); luigi.diserafino@unina.it (L.D.S.); pcirillo@unina.it (P.C.)

**Keywords:** transcatheter aortic valve implantation, staging, cardiac damage, mortality

## Abstract

(1) Aims: We sought to assess the impact of the extent of cardiac damage on survival among real-world patients with severe aortic stenosis (AS) undergoing transcatheter aortic valve implantation (TAVI). (2) Methods: A staging classification was applied to 262 patients from the EffecTAVI Registry at baseline and re-assessed within 30-days after TAVI. The primary endpoint of the study was all-cause mortality at 1-year. Secondary endpoints included cerebrovascular accident, myocardial infarction, permanent pacemaker implantation, endocarditis, and re-hospitalization for all causes. (3) Results: At baseline, 23 (8.7%) patients were in Stage 0/1 (no cardiac damage/left ventricular damage), 106 (40.4%) in Stage 2 (left atrial or mitral valve damage), 59 (22.5%) in Stage 3 (pulmonary vasculature or tricuspid valve damage) and 74 (28.3%) in Stage 4 (right ventricular damage). At 30-days after TAVI, a lower prevalence of advanced stages of cardiac damage than baseline, mainly driven by a significant improvement in left ventricular diastolic parameters and right ventricular function, was reported. At 1-year, a stepwise increase in mortality rates was observed according to staging at baseline: 4.3% in Stage 0/1, 6.6% in Stage 2, 18.6% in Stage 3 and 21.6% in Stage 4 (*p* = 0.08). No differences were found in secondary endpoints. (4) Conclusions: TAVI has an early beneficial impact on the left ventricular diastolic and right ventricular function. However, the extent of cardiac damage at baseline significantly affects the risk of mortality at 1-year after the procedure.

## 1. Introduction

Degenerative aortic stenosis (AS) is the most common heart valve disease among people ≥ 65 years in developed countries, with an increasing prevalence due to population ageing [1,2]. AS commonly leads to left ventricular (LV) pressure overload resulting in concentric hypertrophy that prevents symptom onset for a long time while yielding progressive left and right ventricular dysfunction and impaired survival [3,4]. Current guidelines recommend intervention in patients with AS according to the severity of the disease and the presence of the symptoms [5,6]. However, the benefits of valve replacement may be limited in patients with advanced functional and structural myocardial changes.

A recently proposed staging classification of AS is based on the assumption that there is a continuum in the pathophysiology of LV structural and functional changes induced by AS. Such a system showed prognostic ability among patients from the PARTNER 2 trial as well as in asymptomatic subjects with moderate to severe AS, thus challenging the current management of the disease [7,8].

The aim of our study was to assess the prognostic performance of this staging classification in a real-world cohort of patients undergoing transcatheter aortic valve implantation (TAVI) and to investigate the eventual impact of the procedure on the extent of extra-aortic valve cardiac damage.

## 2. Materials and Methods

### 2.1. Patient Population and Data Collection

All consecutive patients with severe symptomatic AS undergoing TAVI between 2014 and 2019 at our institution were enrolled in the EffecTAVI registry. Severe AS was defined according to current guidelines as a mean aortic valve gradient ≥ 40 mmHg and/or aortic valve area < 1.0 cm^2^ (or an indexed aortic valve area < 0.6 cm^2^/m^2^) and/or a peak aortic jet velocity ≥ 4 m/s [9]. TAVI suitability was established by the local multi-disciplinary Heart Team. All patients in the registry (*n* = 275) received echocardiographic evaluation before TAVI. However, for the present analysis we only considered those with a complete dataset including variables needed for staging cardiac damage according to the system proposed by Gènèreux [7] et al. Clinical, procedural and follow-up data were anonymously entered in a web-based database (https://www.redcap.unina.it). The EffecTAVI registry has been approved by the local ethics committee and written informed consent was obtained for all patients for participation. All study-related procedures were carried out in accordance with the Declaration of Helsinki and this analysis was approved by the institutional review board.

### 2.2. Cardiac Damage Staging Classification

The following criteria for staging classification of cardiac damage were applied at baseline (within 1 month before TAVI) and after the procedure (within 30-days): Stage 0, no extra-aortic valve cardiac damage; Stage 1, LV damage as defined by the presence of LV hypertrophy (LV mass index > 95 g/m^2^ for women, >115 g/m^2^ for men) [10], severe LV diastolic dysfunction (E/e′ > 14) [11] or LV systolic dysfunction (LV ejection fraction, LVEF < 50%); Stage 2, left atrial (LA) and/or mitral valve damage as defined by the presence of LA enlargement (LA volume > 34 mL/m^2^) and/or moderate to severe mitral regurgitation and/or atrial fibrillation; Stage 3, pulmonary vasculature and/or tricuspid valve damage as defined by the presence of systolic pulmonary hypertension (systolic pulmonary arterial pressure, PAPS ≥ 60 mmHg) and/or moderate/severe tricuspid valve regurgitation [12,13]; and Stage 4, right ventricular (RV) damage as defined by the presence of moderate to severe RV systolic dysfunction (tricuspid annular systolic excursion, TAPSE < 17 mm) [10,11,14,15].

To improve the identification of subclinical LV dysfunction, we added the LV global longitudinal strain (GLS) to Stage 1, using a cutoff value of <−20% to define impaired LV-GLS [10]. Patients were hierarchically classified in a given stage (worst stage) if at least 1 of the proposed criteria was met within that stage (Appendix A).

### 2.3. Clinical Follow-Up and Endpoint Assessment

After hospital discharge, follow-up was performed by clinical or phone visits at 30-days and 1-year after TAVI. All adverse events were systematically collected and classified according to the definitions of the Valve Academic Research Consortium-2 [16]. The primary endpoint of the study was all-cause mortality at 1-year after TAVI. Secondary endpoints included cerebrovascular accident, myocardial infarction, permanent pacemaker implantation, endocarditis, and re-hospitalization for all causes.

### 2.4. Statistical Analysis

Continuous data are reported as mean ± standard deviation (SD) and categorical data are reported as frequencies and percentages. Kaplan–Meier curves were constructed to show the rate of mortality up to 1-year across the cardiac damage groups. The association between the extent of cardiac damage and the risk of all-cause mortality at 1-year was assessed by using a Cox-regression multivariable model including all variables associated whose *p*-values were <0.10 in the univariable model. The univariable model was built by using the Student’s *t*-test or Wilcoxon–Mann–Whitney test for continuous variables and the chi-square test or Fisher exact test for categorical variables as appropriate. Statistical analyses were performed with Stata software version 14.2 (StataCorp, College Station, Texas, TX, USA).

## 3. Results

### 3.1. Baseline Characteristics

Among patients included in the EffecTAVI registry between 2014 and 2019, 262 received a detailed echocardiographic assessment at baseline and were included in this analysis. Patients were categorized by the presence and extent of extra-aortic valvular cardiac damage. At baseline, 4 (1.4%) patients were in Stage 0 (no cardiac damage), 19 (7.2%) patients were in Stage 1 (LV damage), 106 (40.4%) patients were in Stage 2 (LA or mitral valve damage), 59 (22.7%) patients were in Stage 3 (pulmonary vasculature or tricuspid valve damage) and 74 (28.3%) patients were in Stage 4 (RV damage). Given the small number of patients in Stage 0 and Stage 1, they were merged in a single group, Stage 0/1. The prevalence of cardiac damage stages and the distribution of their individual components are presented in Figure 1 and Appendix A. The baseline characteristics of the study population according to stage of myocardial damage are provided in Table 1. Patients in more advanced stages of cardiac damage were older and had higher STS score compared with those in lower levels. They also presented with a worse NYHA functional class status and had lower LVEF. Procedural characteristics are summarized in Appendix A. In the majority of cases, TAVI was performed by the transfemoral approach (97.3%) and with self-expanding prosthesis (73.3%). Procedural success according to VARC2 criteria was 98.5% and 6.8% of patients experienced periprocedural complications.

### 3.2. Echocardiographic Assessment at Follow-Up

The echocardiographic evaluation of the overall population is reported in Appendix A. A complete echocardiographic assessment at baseline, discharge and at 30-days after the procedure was available for 130 patients, while 119 did not receive echocardiography at discharge or within 30 days and the evaluation was not complete for 13 patients. Re-assessment of the staging classification yielded more patients in Stage 0/1 and 2 compared with baseline (Table 2 and Figure 2). LVEF showed a marginally, non-significant improvement (55.3 ± 8.1% vs. 55.4 ± 7.3% vs. 58.4 ± 6.6% at baseline, discharge, and 30-days after TAVI; *p* = 0.059). Similarly, GLS significantly improved from baseline to follow-up (16.84 ± 1.8% vs. 19.6% ± 2.3 vs. 19.2 ± 3.5% at baseline, discharge, and 30-days after TAVI; *p* = 0.044). E/e′ ratio, a marker of diastolic dysfunction, showed a peculiar trend: from a mean value of 17.04 ± 4.4 at the baseline, it increased to 20.2 ± 5.7 at discharge and then decreased at 30-days after TAVI with a mean value of 12.19 ± 7.2 (*p* = 0.192). LA volume was significantly reduced (53.8 ± 5.1 mL/m^2^ vs. 52.6 ± 13.9 mL/m^2^ vs. 47.54 ± 11 mL/m^2^ at baseline, discharge, and 30-days after TAVI; *p* < 0.001) as well as systolic pulmonary artery pressure (43.7 ± 8.2 mmHg vs. 41.7 ± 7.4 vs. 37.7 ± 11.1 mmHg, at baseline, discharge, and 30-days after TAVI; *p* < 0.001). Finally, improvement in RV function was demonstrated by an increase in TAPSE (20.46 ± 4.3 mm vs. 20.38 ± 3.7 mm vs. 22.1 ± 5.1, at baseline, discharge, and 30-days after TAVI; *p* = 0.085).

### 3.3. Clinical Outcomes

Clinical outcomes at 1-year after TAVI stratified by stage of cardiac damage are presented in Table 3. All-cause mortality progressively increased from Stage 0/1 (4.3%) to Stage 2 (6.6%), Stage 3 (18.6%) and Stage 4 (21.6%) (*p* = 0.008) (Figure 3). Death for cardiovascular causes occurred in 1 (4.3%) patient in Stage 0/1, 5 (4.7%) patients in Stage 2, 7 (11.9%) patients in Stage 3 and 11 (14.9%) patients in Stage 4 (*p* = 0.012).

No differences were found for secondary endpoints including cerebrovascular accident, myocardial infarction, permanent pacemaker implantation, endocarditis, and re-hospitalization for all causes at 1-year after the procedure. In addition, periprocedural complications did not increase the risk of major clinical outcomes.

In a multivariable model including cardiac damage, NYHA class III/IV, frailty class 4/5, STS score, hypertension, and dyslipidemia, only cardiac damage (HR 1.745, 95% CI 1.17–2.60, *p* = 0.006) and frailty class 4/5 (HR 6.16, 95% CI 1.78–21.37, *p* = 0.004) were significantly associated with all-cause mortality at 1-year follow-up (Appendix A).

## 4. Discussion

The main findings of the present analysis can be summarized as follows:The staging classification of AS-related cardiac changes, derived from a randomized trial, maintains its prognostic performance in real-world TAVI patients;TAVI triggers an early reversal of cardiac dysfunction, mainly driven by the amelioration of LV diastolic and RV function.

Nevertheless, the extent of extra-aortic valve cardiac damage at baseline significantly affects survival at 1-year after the procedure.

The identification of clinical and anatomic factors that affect clinical outcomes of patients with severe AS represents an important unmet need. Several scoring systems that account for baseline features and measures of frailty have been proposed for counselling AS patients [17,18]. However, their use in clinical practice is challenged by the lack or limited availability of all the required variables. In this context, the staging classification of cardiac damage proposed by Généreaux et al. features the unique advantage of being widely applicable as it is based on echocardiographic parameters that are routinely evaluated in patients with severe AS. This system was formulated by leveraging the data of 1661 patients from the PARTNER 2 trial and was proved to be a powerful predictor of mortality at 1-year after aortic valve intervention (either surgical or transcatheter) [7]. In our study, which included real-world patients, the system retains its prognostic ability as a greater extent of cardiac damage was associated with increased risk of all-cause mortality after TAVI. These findings are in line with prior studies that applied the staging classification system in larger populations. In a retrospective analysis of 1189 symptomatic, severe AS patients, stage of cardiac injury was independently associated with all-cause mortality and the combined endpoint of all-cause mortality, stroke, and cardiac-related hospitalization [19]. Among asymptomatic patients with moderate to severe AS, the staging was significantly associated with excess mortality in multivariable analysis adjusted for aortic valve replacement as a time-dependent variable (hazard ratio: 1.31 per each increase in stage; 95% CI: 1.06 to 1.61; *p* = 0.01) and demonstrated incremental value over other traditional risk markers [8]. Another study applied the staging system to TAVI patients and found a graded association between cardiac damage and all-cause mortality [20].

However, our analysis is the first to assess the impact of TAVI on the extent of extra-aortic valve cardiac damage. We found that the procedure triggers an early (within 30-days) re-classification of the stages owing to significant changes in measures of LV diastolic and RV function. LV hypertrophy and collagen abnormalities develop in patients with severe AS and impair diastolic function. Indeed, objective evidence of variable degrees of LV diastolic dysfunction has been reported in up to two-thirds of patients undergoing TAVI [21]. Similarly, RV dysfunction has been documented in up to 1 in 4 patients with severe AS Devereaux a consequence of transmission of elevated left-sided pressure back through the pulmonary vascular system. The suppression of pressure overload by TAVI ameliorates LV filling pressures (E/e′ ratio), as suggested by the concomitant reduction in left atrial volume. Along the same line, a trend towards normalization of TAPSE may occur after TAVI [22,23]. as well as a reduction in pulmonary hypertension [24]. Nevertheless, these changes do not improve survival after TAVI as baseline conditions predominate in determining prognosis at 1-year. We consistently, observed an overall improvement in myocardial function suggested by changes in LV-GLS after TAVI with no relevant impact on mortality at 1-year.

The main clinical implication of the results of our study is the need to rethink the optimal timing of intervention in patients with AS. Multiple lines of evidence indicate that the greater the extent of cardiac damage before TAVI, the higher the probability of worse outcomes after the procedure. Moreover, irreversible structural cardiac changes induced by longstanding AS neutralize the beneficial impact of TAVI on some functional parameters. From this perspective, anticipating the intervention might have the potential advantage of obtaining the full reversibility of cardiac function and improving survival to a greater extent.

Our study has several limitations. It is a retrospective analysis of data collected at a single center; thus, it is subject to inherent flaws related to that design and the sample size may not provide enough power to make definite conclusions. A comprehensive assessment of echocardiographic parameters after TAVI was not performed in the overall population, affecting the completeness of our observations. The parameters used for staging the extent of cardiac damage are those obtained in the context of the routine echocardiogram of TAVI patients and are potentially subject to measurement errors and variability. We did not assess the potential modifying effect of paravalvular aortic regurgitation on echocardiographic and clinical outcomes after TAVI. Moreover, we could not evaluate whether adding cardiac biomarkers might ameliorate the prognostic role of the staging classification as they are not routinely measured in our clinical practice.

## 5. Conclusions

The staging classification confirms its utility as an additive clinical tool to enhance risk stratification and therapeutic decision making in patients with AS. Timely intervention (either transcatheter or surgical) might reverse functional cardiac changes associated with AS; however, survival at medium-term is mainly related to the baseline grade of the extent of cardiac damage. Further and larger studies are needed to assess the value of the staging classification in the post-procedural setting.

## Figures and Tables

**Figure 1 jcm-10-04563-f001:**
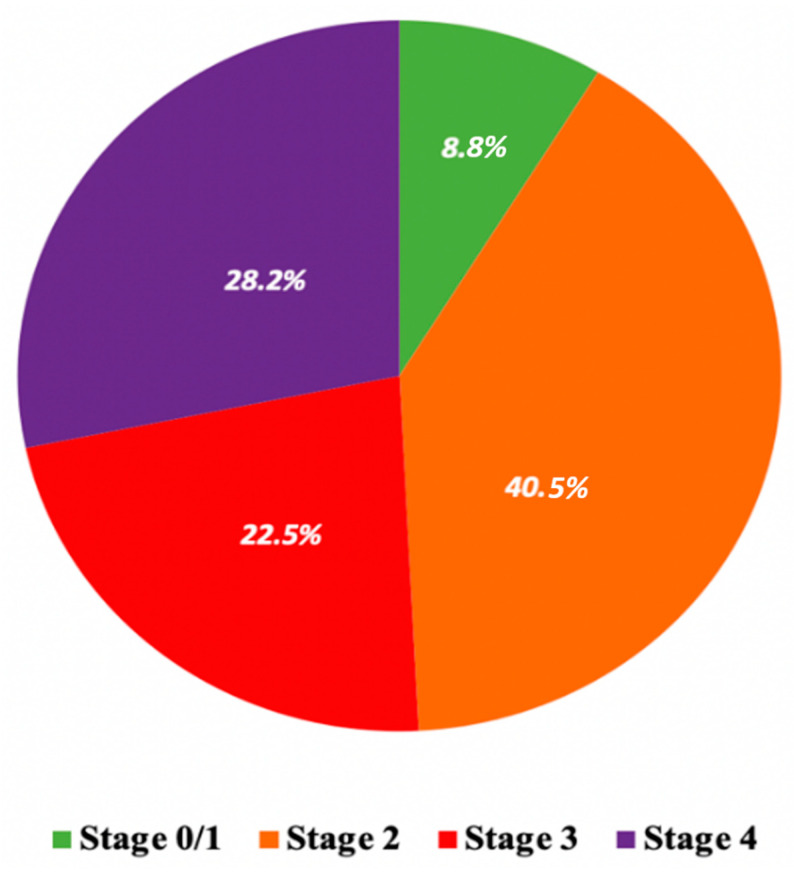
Distribution of stages of cardiac damage in the study population.

**Figure 2 jcm-10-04563-f002:**
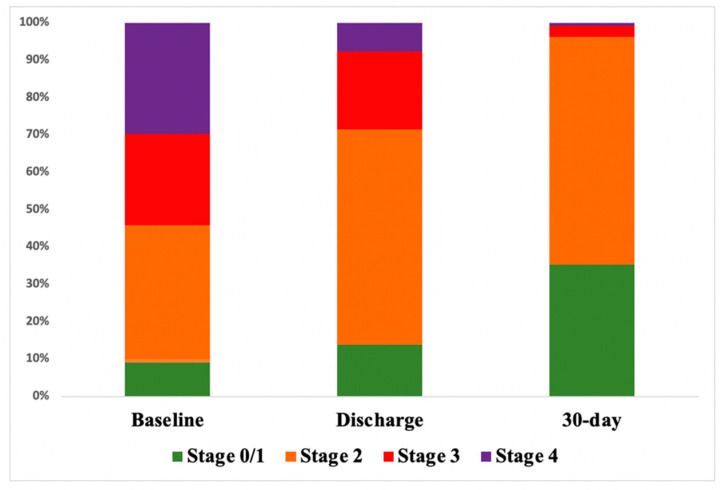
Re-assessment of cardiac damage at discharge and at 30-days after TAVI.

**Figure 3 jcm-10-04563-f003:**
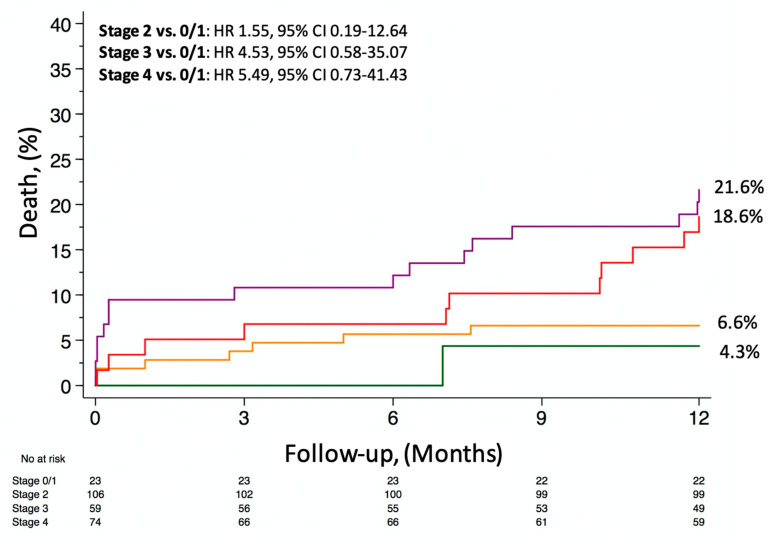
All-cause mortality according to the stage of cardiac damage.

**Table 1 jcm-10-04563-t001:** Baseline characteristics of patient population according to the stage of cardiac damage.

	Stage 0/1 (*n* = 23)	Stage 2 (*n* = 106)	Stage 3 (*n* = 59)	Stage 4 (*n* = 74)
Age, years	77.3 ± 6.8	79.2 ± 6.7	81.1 ± 5.2	80.4 ± 5.7
Female sex	14 (60.9%)	67 (63.2%)	41 (69.5%)	40 (54.1%)
BMI (kg/m^2^)	26.4 ± 4.3	27.1 ± 5.9	27.3 ± 5.7	28.1 ± 6
Hypertension	19 (82.6%)	94 (88.7%)	51 (86.4%)	65 (87.8%)
Diabetes mellitus	5 (21.7%)	41 (38.7%)	16 (27.1%)	25 (33.8%)
Dyslipidemia	12 (52.2%)	69 (65.1%)	34 (57.6%)	44 (59.5%)
Coronary artery disease	7 (30.4%)	49 (46.2%)	25 (42.4%)	36 (48.6%)
Previous myocardial infarction	1 (4.3%)	18 (17%)	12 (20.3%)	16 (21.6%)
Previous cerebrovascular accident	2 (8.7%)	11 (10.4%)	3 (5.1%)	12 (16.2%)
Peripheral artery disease	11 (47.8%)	57 (53.8%)	23 (39%)	45 (60.8%)
Chronic kidney disease	5 (21.7%)	34 (32.1%)	15 (25.4%)	21 (28.4%)
COPD	7 (30.4%)	26 (24.5%)	11 (18.6%)	27 (36.5%)
Dyspnea	11 (47.8%)	84 (79.2%)	43 (72.9%)	64 (86.5%)
Angina	5 (21.7%)	27 (25.5%)	16 (27.1%)	20 (27%)
Syncope	5 (21.7%)	15 (14.2%)	6 (10.2%)	7 (9.5%)
LVEF (%)	60.5 ± 8.9	55.9 ± 9.6	54.5 ± 11.7	49.3 ± 13.1
STS-PROM score (%)	3.5 ± 1.4	4.7 ± 3.1	5.2 ± 3.2	6.6 ± 4.8
Frailty scale				
0–1, *n* (%)	11 (47.8%)	41 (38.6%)	22 (37.3%)	21 (28.4%)
2–3, *n* (%)	12 (52.2%)	60 (56.6%)	36 (61%)	52 (70.3%)
4–5, *n* (%)	0 (0%)	5 (4.7%)	1 (1.7%)	1 (1.4%)
NYHA functional class				
I or II, *n* (%)	18 (78.3%)	48 (45.3%)	25(42.4%)	20 (27%)
III or IV, *n* (%)	5 (21.7%)	58 (54.7%)	34(57.6%)	54 (73%)

**Table 2 jcm-10-04563-t002:** Re-assessment of cardiac damage at discharge and 30-days after TAVI.

	Baseline	Discharge	30-Days
Stage 0/1	12/130 (9.2%)	18/130 (13.8%)	46/130 (35.4%)
Stage 2	47/130 (36.2%)	75/130 (57.7%)	79/130 (60.8%)
Stage 3	32/130 (24.6%)	27/130 (20.9%)	4/130 (3.1%)
Stage 4	39/130 (30%)	10/130 (7.7%)	1/130 (0.7%)
LVEF,%	55.3 ± 8.1	55.4 ± 7.3	58.4 ± 6.6
GLS,%	−6.84 ± 1.8	−19.6 ± 2.3	−19.2 ± 3.5
E/e′ ratio	17.04 ± 4.4	20.2 ± 5.7	12.09 ± 7.2
LAVI, mL/m^2^	53.83 ± 5.1	52.6 ± 13.9	47.54 ± 11
PAPS, mmHg	43.7 ± 8.2	41.7 ± 7.4	37.7 ± 11.1
TAPSE, mm	20.46 ± 4.3	20.38 ± 3.7	22.04 ± 5.1

**Table 3 jcm-10-04563-t003:** Clinical outcomes at 1-year after TAVI stratified by stage of cardiac damage.

	Stage 0/1 (*n* = 23)	Stage 2 (*n* = 106)	Stage 3 (*n* = 59)	Stage 4 (*n* = 74)	*p*-Value
All-cause death	1 (4.3%)	7 (6.6%)	10 (16.7%)	15 (20.3%)	0.008
Cardiovascular death	1 (4.3%)	5 (4.7%)	7 (11.9%)	11 (14.9%)	0.012
Stroke	0 (0%)	2 (1.9%)	2 (3.4%)	3 (4.1%)	0.67
Myocardial infarction	0 (0%)	2 (1.9%)	2 (3.4%)	5 (6.8%)	0.25
Permanent pacemaker implantation	6 (26.1%)	15(14.2%)	7 (11.9%)	16 (21.6%)	0.24
Endocarditis	1 (4.3%)	1 (0.9%)	0 (0%)	2 (2.7%)	0.38
Re-hospitalization	0 (0%)	10 (9.4%)	4 (6.8%)	9 (12.2%)	0.79

## Data Availability

The data presented in this study are available on request from the corresponding author. The data are not publicly available due to firewalls to access our dataset from outside our Institution.

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
