# Peer review of "Extent of Cardiac Damage and Mortality in Patients Undergoing Transcatheter Aortic Valve Implantation"

_jcm, 2021, doi:10.3390/jcm10194563_

Round 1

Reviewer 1 Report

The paper highlights the central role played by extra-aortic cardiac damage on the mean term outcome for patients treated with TAVI for degenerative aortic stenosis. This finding was previously shown only in randomized trials however according to the authors the same trend is confermed.

These are my doubts note:

1) Speaking of the statistical analysis comparing different groups of patients, (TAB 1) baseline characteristic and (TAB2) re-assessment of cardiac damage at disccharge, it's not clear to me which group of patients the P value refers to. Did you consider performing a global variance analysis or the variance between groups?

2) Any biomarker values are referred, could you consider to implement it for a better risk stratification?

3) A complete echo assessment is obteined only in 130 patients out of 262 baseline patients. How many of these are assessed at discharge and how many at 30-days?

4) 6.8% of the patients experienced periprocedural complications, is there any relationship berween early complication and cardiac damage?

Author Response

List of Revisions and Comments in response to Reviewer’s Requests

(Reviewer comments in bold; response in plain text; revised pieces of text are shown in italics)

Reviewer #1

The paper highlights the central role played by extra-aortic cardiac damage on the mean term outcome for patients treated with TAVI for degenerative aortic stenosis. This finding was previously shown only in randomized trials however according to the authors the same trend is confirmed.

We thank the Reviewer for the careful and thoughtful review of our manuscript.

  1. Speaking of the statistical analysis comparing different groups of patients, (TAB 1) baseline characteristic and (TAB2) re-assessment of cardiac damage at discharge, it's not clear to me which group of patients the P value refers to. Did you consider performing a global variance analysis or the variance between groups?

We are grateful for this comment. The actual presentation of the results in Table 1 and 2 is, indeed, misleading. These tables are meant to be only descriptive whereas it was not our intention to show a comparison of clinical and echocardiographic features among groups and, in the revised version of the manuscript, p-values are not reported. Statistical analysis section was changed accordingly:

  • Material and Methods Section: “Continuous data are reported as mean ± standard deviation SD and categorical data are reported as frequencies and percentages. Kaplan-Meier curves were constructed to show the rate of mortality up to 1-year across the cardiac damage groups. The association between the extent of cardiac damage and the risk of all-cause mortality at 1-year was assessed by using a Cox-regression multivariable model including all variables associated whose p-values were <0.10 at univariable model. The univariable model was built by using the Student’s t-test or Wilcoxon-Mann-Whitney test for continuous variables and the chi-square test or Fisher exact test for categorical variables as appropriate. Statistical analyses were performed with Stata software version 14.2 (StataCorp, College Station, Texas).” (Page 3, Lines 108-116).
  1. Any biomarker values are referred, could you consider to implement it for a better risk stratification?

We thank the Reviewer for raising this point. Biomarkers may play a very important role in the risk stratification of patients with structural heart disease. However, our registry does not allow an accurate evaluation of the potential prognostic role of cardiac biomarkers as pro-BNP and galectin-3 levels at baseline and follow-up were available for only 71 and 67 patients, respectively. We acknowledged this limitation in the appropriate section, as follows:

  • Discussion section: “Moreover, we could not evaluate whether adding cardiac biomarkers might ameliorate the prognostic role of the staging classification as they are not routinely measured in our clinical practice.” (Page 8, Lines 269-271).
  1. A complete echo assessment is obtained only in 130 patients out of 262 baseline patients. How many of these are assessed at discharge and how many at 30-days?

Thank you for this comment. Of the 130 patients included in the re-assessment of cardiac damage after TAVI, all received complete echocardiographic evaluation at baseline, discharge and 30-day after TAVI. On the other side, 132 patients were excluded because of the following reasons: 119 did not receive echocardiography at discharge or within 30 days (of these, 10 subjects died within 1 month after the procedure) and evaluation was not complete for 13 patients (missing variables). In the revised paper, we report the following:

  • Results section, Echocardiographic assessment at follow-up: “119 did not receive echocardiography at discharge or within 30 days and evaluation was not complete for 13 patients.” (Page 5, Lines 165-166).
  1. 8% of the patients experienced periprocedural complications, is there any relationship between early complication and cardiac damage?

We thank the Reviewer for this comment. We did not observe a significant difference in the rate of periprocedural complications among the different stages of cardiac damage.

We included the following in the revised version of the manuscript:

  • Results section: “In addition, periprocedural complications did not increase the risk of major clinical outcomes.” (Page 6, Lines 195-196).

Reviewer 2 Report

Dear Editor and Authors,

I was asked to review the manuscript titled “Extent of cardiac damage and mortality in patients undergoing transcatheter aortic valve implantation” by Dr. Avvedimento and colleagues from the University of Naples Federico II in Italy.

In this retrospective analysis the authors utilized data from their institution entered into the EffecTAVI Registry to assess the impact of the extent of cardiac damage on survival among patients with severe aortic stenosis (AS) who underwent  transcatheter aortic valve implan-tation (TAVI).

The manuscript is well written and presented, in understandable English and it is easy to read. There are a number of issues that I feel need addressing regarding this work.

Specifically:

Major issues:

  1. How complete were the data in the registry? Where there missing variables? Are regular checks/audits performed to ensure data completeness and accuracy?
  2. The sample size of the patients studied seems adequate at 262 patients however no pre-analysis sample size calculation or power analysis was performed to confirm this feeling!! Why wasn’t a sample size calculation done at the design phase of the study?
  3. Is the cardiac classification system the authors have utilized validated? It is based on a previously published classification system by Généreux, et al. but I believe it is not broadly used in clinical practice! The PARTNER trial included also surgical AVR patients (547/1114) therefore there is a question of validity/applicability.
  4. Why wasn’t a multivariable analysis performed to assess which factors were significantly correlated with each cardiac damage stage? (Table 1) This would have been quite easy and would provide statistically more robust information!
  5. Only 130 patients had a complete pre-procedure and post-procedure echocardiogram. This as the authors have correctly stated is a limitation of their study as the power of their analysis is reduced!
  6. Similarly to 4, why wasn’t a multivariable analysis performed to assess the variables associated with cardiac damage at 1-year?

Minor issues:

  1. Line 58. The authors state “Patients with a complete echocardiographic evaluation at baseline were con-58 sidered eligible for the present analysis.” You mean that there were patients undergoing a major procedure such as TAVI without a complete echocardiographic evaluation!! This statement is redundant and discourteous to the audience!
  2. Line 61. The authors state “The EffecTAVI registry has been approved by the local ethic committee and all study related procedures were carried out in accordance with the Declaration of Helsinki. Written informed consent was obtained for all patients for participation in this registry.” Nowhere do they mention that this specific analysis was approved by the local ethics committee!! Was approval sought and obtained? Informed consent by each patient is not required since they have agreed to their date been collected and utilized for research and analysis BUT each analysis/project needs to have ethical approval!
  3. Line 109. What is “fighle” is this a typo? Did they mean table?
  1. It is customary in tables to give along with the number of patients (n) the percentage (%) of the total. This is not done in table 2 in the fist 4 lines (cardiac stages).
  2. The main message the authors are understandably given their specialty, eagerly trying to put forward is that “we need to rethink the optimal timing of intervention in patients with AS as the greater the extent of cardiac damage before TAVI, the higher the probability of worse outcomes after the procedure.” However, would that not change the indications for surgery as well? TAVI is promoted as a salvage procedure in patients who are unable to safely undergo surgical AVR. If we were to push for an earlier intervention in patients (before suffering serious cardiac damage and becoming unable to have surgery) would that not infringe upon the surgical patient population who proven will see a much better long term benefit with a open procedure AVR?   None of the authors of the manuscript appear to be a cardiac surgeon (despite the fact they are an integral and equal part of the Heart Team) and thus these conclusions in the discussion appear to be cardiology perspective biased!

In conclusion, this is an interesting analysis but it is my feeling it needs some additional work and modification before it is ready to be presented in the literature. Thank you again and I wish good luck and health to all.

Author Response

List of Revisions and Comments in response to Reviewer’s Requests

(Reviewer comments in bold; response in plain text; revised pieces of text are shown in italics)

Reviewer #2

Dear Editor and Authors,

I was asked to review the manuscript titled “Extent of cardiac damage and mortality in patients undergoing transcatheter aortic valve implantation” by Dr. Avvedimento and colleagues from the University of Naples Federico II in Italy.

In this retrospective analysis the authors utilized data from their institution entered into the EffecTAVI Registry to assess the impact of the extent of cardiac damage on survival among patients with severe aortic stenosis (AS) who underwent transcatheter aortic valve implantation (TAVI).

The manuscript is well written and presented, in understandable English and it is easy to read.

We thank the Reviewer for her/his detailed review of our manuscript.

There are a number of issues that I feel need addressing regarding this work.

Major issues:

  1. How complete were the data in the registry? Where there missing variables? Are regular checks/audits performed to ensure data completeness and accuracy?

We thank the Reviewer for this comment. Although a regular audit is locally performed once monthly to ensure data completeness and accuracy, not all variables in our registry are available for all patients. This happens mainly because of logistic reasons: for example, challenges for patients to come to the hospital for follow-up visits, echocardiographic evaluation performed in other hospitals, etc. In the revised manuscript, we provided details about missing variables:

  • Results section, Echocardiographic assessment at follow-up: “119 did not receive echocardiography at discharge or within 30 days and evaluation was not complete for 13 patients.” (Page 5, Lines 165-166).
  1. The sample size of the patients studied seems adequate at 262 patients however no pre-analysis sample size calculation or power analysis was performed to confirm this feeling!! Why wasn't a sample size calculation done at the design phase of the study?

This is a relevant point. Our main objective was to assess the prognostic performance of the staging classification system proposed by Gènèreux et al in a real-world cohort of TAVI patients. We did not calculate the sample size because our study is characterized as a descriptive one. In addition, it seems not appropriate to make a formal post-hoc estimation of the population size. We concur with the Reviewer that this represents a further limitation of our work that we acknowledged as follows:

  • Discussion section: “It is a retrospective analysis of data collected at a single center, thus subject to inherent flaws related to that design and whose sample size may be not powered to make definite conclusions.” (Page 8, Lines 261-263).
  1. Is the cardiac classification system the authors have utilized validated? It is based on a previously published classification system by Gènèreux, et al. but I believe it is not broadly used in clinical practice! The PARTNER trial included also surgical AVR patients (547/1114) therefore there is a question of validity/applicability.

We concur with the Reviewer that the staging system by Gènèreux et al. is not currently used in clinical practice. However, we do see a realistic application of this tool in the next future as the optimal timing of intervention in patients with aortic stenosis is still to be determined. We highlighted the potential role of the system in the Discussion section (The main clinical implication of the results of our study is the need for rethinking the optimal timing of intervention in patients with AS. Multiple lines of evidence indicate that the greater is the extent of cardiac damage before TAVI, the higher is the probability of worse outcomes after the procedure. Moreover, irreversible structural cardiac changes induced by longstanding AS neutralize the beneficial impact of TAVI on some functional parameters. In this perspective, anticipating the intervention might have the potential advantage to obtain the full reversibility of cardiac function and improve survival at a greater extent.). Several other studies assessed the prognostic performance of the staging system in its original version or with some changes among different populations of patients with aortic stenosis (undergoing TAVI, SAVR or asymptomatic). As such, we do believe that our study corroborates prior evidence that the system is widely applicable to TAVI patients.

  1. Why wasn't a multivariable analysis performed to assess which factors were significantly correlated with each cardiac damage stage? (Table 1) This would have been quite easy and would provide statistically more robust information!

The aim of our study was to assess the impact of cardiac damage stage at baseline on clinical outcomes at 1-year. We simply applied the staging system, as it was proposed by Gènèreux et al.  to our population. Contrariwise and based on the comment 6 of this Reviewer, we performed a multivariable analysis to identify factors associated with mortality at 1-year.

  1. Only 130 patients had a complete pre-procedure and post-procedure echocardiogram. This as the authors have correctly stated is a limitation of their study as the power of their analysis is reduced!

This represents a major limitation as stated by the Reviewer and reported in the limitation section. Moreover, we detailed reasons for data incompleteness:

  • Results section, Echocardiographic assessment at follow-up: “119 did not receive echocardiography at discharge or within 30 days and evaluation was not complete for 13 patients.” (Page 5, Lines 165-166).
  1. Similarly to 4, why wasn't a multivariable analysis performed to assess the variables associated with cardiac damage at 1-year?

We thank the Reviewer for raising this very important point. Actually, we did not assess cardiac damage at 1-year but the impact of staging at baseline on clinical outcomes at 1-year.  Based on this comment, we performed a multivariable analysis and we found that cardiac damage at baseline is independently associated with all-cause death at 1-year. The manuscript was adapted as follows:

  • Materials and Methods Section: “Continuous data are reported as mean ± standard deviation SD and categorical data are reported as frequencies and percentages. Kaplan-Meier curves were constructed to show the rate of mortality up to 1-year across the cardiac damage groups. The association between the extent of cardiac damage and the risk of all-cause mortality at 1-year was assessed by using a Cox-regression multivariable model including all variables associated whose p-values were <0.10 at univariable model. The univariable model was built by using the Student’s t-test or Wilcoxon-Mann-Whitney test for continuous variables and the chi-square test or Fisher exact test for categorical variables as appropriate. Statistical analyses were performed with Stata software version 14.2 (StataCorp, College Station, Texas).” (Page 3, Lines 108-116).
  • Results Section: “In a multivariable model including cardiac damage, NYHA class III/IV, frailty class 4/5, STS score, hypertension, and dyslipidemia, only cardiac damage (HR 1.745, 95%CI 1.17-2.60, P=0.006) and frailty 4/5 class (HR 6.16, 95%CI 1.78-21.37, P=0.004) were significantly associated with all-cause mortality at 1-year follow-up (Supplemental Table 4).” (Page 6, Lines 197-200)

Supplemental Table 4. Multivariable model 

Hazard Ratio

Standard error

95% Confidence Intervals

p- value

Stage of cardiac damage

1.75

0.35

1.17-2.60

0.006

NYHA class

1.73

0.67

0.81-3.71

0.159

Fraility

6.16

3.90

1.77-21.37

0.004

STS score

1.04

0.03

0.97-1.11

0.231

Hypertension

2.58

6.55

0

1.0

Dylipidemia

1.53

0.60

0.71-3.31

0.283

Round 2

Reviewer 2 Report

Dear Editor and Authors,

I have re-read and evaluated the submitted revised manuscript and the comments response of the authors. Although I still have some reservations regarding the applicability and validity of the findings in terms of power of analysis and also in regards to the methodology of the study, I feel that the authors have attempted to honestly address the limitations of their study and have responded and made edits to the comments made by the reviewers adequately. Therefore, I am now agreeing to the publication of this work. Congratulations to the authors, I hope in their next analysis/project they consider the comments made by myself, which I feel they understand were made with constructive and educational intent, so that they can design a more robust methodologically design. I will well to all.